# An Alkyne-Mediated SERS Aptasensor for Anti-Interference Ochratoxin A Detection in Real Samples

**DOI:** 10.3390/foods11213407

**Published:** 2022-10-28

**Authors:** Hao Wang, Lu Chen, Min Li, Yongxin She, Chao Zhu, Mengmeng Yan

**Affiliations:** 1Institute of Quality Standard and Testing Technology for Agro-Products, Shandong Academy of Agricultural Sciences, Jinan 250100, China; 2Shandong Provincial Key Laboratory Test Technology on Food Quality and Safety, Jinan 250100, China; 3Institute of Quality Standard and Testing Technology for Agro-Products, Chinese Academy of Agricultural Science, Beijing 100081, China

**Keywords:** SERS, 4-TEAE, aptamer, silent region, OTA, anti-interference

## Abstract

Avoiding interference and realizing the precise detection of mycotoxins in complex food samples is still an urgent problem for surface-enhanced Raman spectroscopy (SERS) analysis technology. Herein, a highly sensitive and specific aptasensor was developed for the anti-interference detection of Ochratoxin A (OTA). In this aptasensor, 4-[(Trimethylsilyl) ethynyl] aniline was employed as an anti-interference Raman reporter to prove a sharp Raman peak (1998 cm^−1^) in silent region, which could avoid the interference of food bio-molecules in 600–1800 cm^−1^. 4-TEAE and OTA-aptamer were assembled on Au NPs to serve as anti-interference SERS probes. Meanwhile, Fe_3_O_4_ NPs, linked with complementary aptamer (cApts), were applied as capture probes. The specific binding of OTA to aptamer hindered the complementary binding of aptamer and cApt, which inhibited the binding of SERS probes and capture probes. Hence, the Raman responses at 1998 cm^−1^ were negatively correlated with the OTA level. Under the optimum condition, the aptasensor presented a linear response for OTA detection in the range of 0.1–40 nM, with low detection limits of 30 pM. In addition, the aptasensor was successfully applied to quantify OTA in soybean, grape and milk samples. Accordingly, this anti-interference aptasensor could perform specific, sensitive and precise detection of OTA in real samples, and proved a reliable reference strategy for other small-molecules detection in food samples.

## 1. Introduction

Ochratoxin A (OTA) is a naturally occurring mycotoxin that is one of the most toxic contaminants in dairy products, crops and alcoholic beverages, with a long half-life and high chemical stability [1]. OTA has nephrotoxicity, hepatotoxicity, and immunotoxicity for humans; hence, many countries have regulations for OTA limits. According to European Commission regulations, the maximum level for OTA in grains, grape juice and wine are 5 μg/kg, 2 μg/kg and 2 μg/kg, respectively [2]. To avoid the hazards of OTA for human health, quantifying the contaminated level of OTA in agricultural products is of great significance. In recent decades, instrumental methods [3] and antibody-based methods [4] have developed to be prevalent methods for OTA detection. However, expensive instruments and the demand for trained technicians limit the development of instrumental methods. In addition, high prices and the stringent conditions of transportation and storage limit the practical applicability of antibody-based methods [5]. Therefore, it is highly desirable to explore simple, low-cost and sensitive OTA detection methods.

In recent years, surface-enhanced Raman spectroscopy (SERS)-based biosensor has acted as a promising alternative in food safety, due to its high sensitivity, specificity and multiplexing detection ability [3,6]. As the core component of SERS-based biosensor, SERS probes are normally composed of metal nanostructures (enhanced substrates), such as gold nanoparticles (Au NPs) and Raman reporters (RRs) [7]. RRs (such as 4-mercaptobenzoic acid, 4-mercaptopyridine, 4-nitrothiophenol, etc.) are easily and firmly attached to the surface of metal nanostructures, which could enhance the inherent Raman emissions of RRs to 10^6^–10^14^ times under the action of electromagnetic mechanism and chemical enhancement [8]. SERS probes provide clear, strong and steady Raman signals of RRs, which could perform high sensitivity and high fidelity analysis [9]. In recent years, some biosensors based on SERS probes for the quantitative analysis of mycotoxins (such as zearalenone and aflatoxin B1) [10,11] have been developed. However, there are still some problems that hindered the practical application of SERS-based biosensors in food analysis. In particular, the Raman emission of food biomolecules and traditional RRs are formed in the region of <1800 cm^−1^; resulting Raman interference could seriously affect the accuracy of the detection results. Therefore, it is urgent to develop an anti-interference RRs for food analysis [12,13].

Recently, researchers have reported some new RRs, which contain C≡C, C=O and C≡N groups [14,15,16]. Those RRs displayed narrow peak in the silent region (1800–2800 cm^−1^), which can effectively avoid the Raman interference of food biomolecules. As new RRs are created, those anti-interference RRs have been applied in SERS technology, such as biosensors, cell imaging, biomedicine and other fields [17,18,19,20]. As a major classification of anti-interference RRs, alkynyl-containing RRs have unique Raman shift in the silent region. The exploration of alkynyl-containing RRs is helpful to the development of Raman silent region tags and expand the application potential of anti-interference RRs. However, thus far, there are no reports about applying alkynyl compound as anti-interference RRs in the food safety field.

Therefore, we present an alkyne-mediated SERS-based aptasensor for OTA detection in real samples (soybean, grape and milk). As shown in Figure 1, 4-[(Trimethylsilyl) ethynyl] aniline (4-TEAE) was applied as anti-interference Raman reporter; gold nanoparticles (Au NPs) linked with aptamers and 4-TEAE were used as Raman probes (4-TEAE/AuNPs/Apt). Meanwhile, Fe_3_O_4_ nanoparticles (Fe_3_O_4_ NPs) conjugated with cApt served as capture probes (Fe_3_O_4_ NPs/cApt). OTA could specifically interact with aptamer, which inhibited the conjunction between Raman probes and capture probes. Thus, the Raman intensity of 4-TEAE/AuNPs was inverse, correlating with the concentration of OTA. In order to verify the practicability of the aptasensor, we detected OTA in soybean, grape and milk samples, and compared the recovery rates with HPLC-MS/MS.

## 2. Materials and Methods

### 2.1. Materials and Reagent

Ochratoxin A, chloroauric Acid (HAuCl_4_), trisodium citrate, 4-[(Trimethylsilyl) ethynyl] aniline, Tris (2-carboxyethy1) phosphine (TCEP), phosphate buffer saline (PBS), ethyl-3-(3-dimethylaminopropyl) carbodiimide hydrochloride (EDC), and N-Hydroxysuccinimide (NHS) were purchased from Aladdin Industrial Inc (Shanghai, China). Fe_3_O_4_ nanoparticles bonded with -COOH (10 mg/mL) were bought from Sigma-Aldrich (Beijing, China).

According to previous research [21], the sequences of aptamer and cApt are as follows: 5′-SH-GATCGGGTGTGGGTGGCGTAAAGGGAGCATCGGACA-3′ (aptamer), 5′-NH2-(CH_2_)_6_-CCTTTACGCCACCCACACCCGATC-3′ (cApt). All synthetic nucleotide sequences were acquired from Sangon Biotech Co., Ltd. (Shanghai, China)

### 2.2. Instruments

The synthesis of Au NPs was carried out by Magnetic electric heating sleeve (IKA, staufen, Germany). The recording and comparison of Raman spectrums were performed by miniature handheld Raman spectrometer (QEPro, Ocean Insight, Dunedin, FL, USA), and the laser power and laser wavelength of spectrometer were 390-410 mW and 785 nm, respectively. UV-vis spectra (synergy HTX, Biotek, Winooski, VT, USA) was applied to get ultraviolet visible (UV) absorbance. The morphology of nanoparticles were characterized by Transmission electron microscope (TEM) (JEM1200EX, JEOL, TKY, Nagoya, Japan). The chemical composition and relative content of the probes were obtained by the energy-dispersive X-ray (EDX) spectroscopy facility of scanning electron microscopy (SEM) (SU8020, Hitachi, TKY, Japan). Ultra-pure water was prepared by purification system (Millipore, Bedford, MA, USA) and used throughout this work. 

### 2.3. Preparation of SERS Probe

Au NPs (25 nm) were synthesized by the sodium citrate reduction method [22]. First, 100 mL ultra-pure water was heated to boiling with a heating sleeve; 1 mL (0.01 g/mL) trisodium citrate solution was then added. After 1 min, 0.1 mL (0.1 g/mL) chloroauric acid solution was added to the boiling solution, and the color of the boiling liquid quickly changed from colorless to transparent wine red. After the color of Au NPs solution was stable and no longer changed, heating ceased and continuous stirring was performed for 20 min. After naturally cooling to room temperature, the Au NPs solution was stored at 4 °C for further use.

OTA aptamer was dissolved to 100 μM with TCEP solution and then diluted to 10 μM with PBS solution for further use. The 0.87 mL Au NPs solution was mixed with 100 μL 4-TEAE in a scroll oscillator for 1 h. The resulted solution was centrifuged under 6000 rpm for 15 min and washed with ultra-pure water three times to obtain 4-TEAE fixed Au NPs (Au NPs-4-TEAE). Then, 30 μL (10 μM) OTA aptamer was co-incubated with Au NPs-4-TEAE solution at room temperature for 4 h. Finally, the Au NPs-4-TEAE/Apt solution was centrifuged under 6000 rpm for 15 min, washed with ultra-pure water three times and re-dissolved in PBS solution to obtain anti-biomolecule interference SERS probes, and stored at 4 °C.

### 2.4. Preparation of Capture Probes

The surface of MNPs was immobilized with cApt and the capture probes were prepared. Briefly, 100 μL MNPs (10 mg/mL) solution was mixed with 7.9 mL ultra-pure water; 1 mL EDC (16 mM) and 1 mL NHS (4 mM) were added to protect the carboxyl groups on the surface of MNPs. After 30 min, MNPs solution was separated by magnet, washed with ultra-pure water three times, and then redissolved to 9.7 mL with PBS solution. Subsequently, 0.3 mL of cApt (10 μM) solution was added to the carboxyl-protected MNPs solution and vibrated for 1 h to prepare capture probes. After magnetic separation and removing the excess cApt, the capture probes were re-dissolved by PBS solution and stored at 4 °C for subsequent use.

### 2.5. Interference-Free Aptasensors for OTA Detection

Firstly, 1 mg/mL OTA acetonitrile solution was diluted with ultra-pure to 0.1–70 nM for further use. Secondly, a competitive binding system was constructed to link the OTA concentration with the SERS intensity at 1998 cm^−1^. Briefly, 500 μL anti-interference SERS probe and 500 μL capture probe were mixed for 15 min, and then 20 μL OTA solution of different concentrations were added to the mixed solution and incubated with vibration for 25 min to fully react. Finally, the SERS probes that were bound to the capture probes were removed by magnetic separation and washed with ultra-pure water two times, and the remaining probes were suspended in the 1 mL of ultra-pure water for SERS measurement.

Finally, the SERS response at different OTA concentrations was measured by miniature handheld Raman spectrometer (laser wavelength = 785 nm and spectrometer power = 390–410 mW), and the response results to different OTA concentrations were obtained by the statistical results of three parallel measurements.

### 2.6. Procedures for OTA Residues Detection in Real Samples

To verify the practicability of the aptasensor, soybean, grape and milk were purchased from the supermarket and tested as real food samples. The pretreatment methods of food samples were referred to in previous reports [23,24,25]. The actual samples were added 1 nM, 5 nM and 10 nM of OTA, respectively. The extracted samples were added to the aptasensor system to complete the detection.

### 2.7. Specificity Analysis

In this work, AFB1, aflatoxin M1 (AFM1), zearalenone (ZEN), deoxynivalenol (DON) and trichosporene 2 (TMAE 2) were selected to verify the specificity of the aptasensor. Under the same conditions, these mycotoxins of 300 nM were simultaneously detected by the aptasensor with the OTA of 30 nM. The assays were repeated three times, and the SERS intensities at 1998 cm^−1^ were compared.

### 2.8. Statistical Analysis

All results were obtained from cubic parallel experiments, and standard deviation (SD) were represented with error bars one. Statistical analysis was performed with Origin 9.5. Ultimately, the limit of detection (LOD) was computed as LOD = 3 × SD/blank.

## 3. Results and Discussion

### 3.1. Synthesis and Characterization of 4-TEAE/Au NPs

In this study, 4-TEAE was applied as anti-interference Raman reporter to prove a strong and stable Raman signal at 1998 cm^−1^, with the following reasoning: (1) Alkynyl linking with benzene ring presents sharp Raman scattering as the large Raman cross section of aromatic ring [26]. (2) 4-TEAE possesses a distinct Raman scattering at 1998 cm^−1^ due to the stretch of C≡C bond, achieving the anti-interference OTA detection in the complex samples. The Raman shifts of other Raman peaks of 4-TEAE, such as 1601 cm^−1^ and 1174 cm^−1^, were not adopted because of the interference of the food biomolecules in the range of <1800 cm^−1^ (Figure 2d).

The sharp Ramam scattering in 1998 cm^−1^ contributed to the combination of SERS enhanced substrate (Au NPs) and Raman reporter (4-TEAE). TEM was applied to image the combination of 4-TEAE/Au NPs. The synthesized Au NPs were spherical, with a particle size of 25 nm (Appendix A). After linked with 4-TEAE, the nanoparticles were wrapped by a shadow with the thickness of 1.5 nm (Appendix A). The combination of Au NPs and 4-TEAE was further identified by UV–vis spectra and EDX. UV–vis spectra confirmed that the absorption peak of Au NPs was red shifted after being coated by 4-TEAE. As shown in Figure 2c, the absorption peak of Au NPs presented at 524 nm, while the absorption peak of 4-TEAE/Au NPs was located at 526 nm. As revealed by Figure 2a,b, the EDX spectroscopy of Au NPs only possessed the peaks of Au, while the Au, C and N elements were co-existent in the 4-TEAE/Au NPs sample.

In order to verify the anti-interference ability of 4-TEAE, three traditional RRs (2,2′-Bipyridyl, Rhodamine 6G, 4-mercaptopyridine) were selected as Interfered control. The Raman scatterings of these RRs and food biomolecules were overlapping in the range of <1800 cm^−1^. While 4-TEAE presented a distinct Raman signal in the silent region. 2,2′-Bipyridyl was chosen as the representative of traditional RRs and tested for the anti-interference ability of 4-TEAE. As shown in Figure 2d,e, the characteristic peaks of 2,2′-Bipyridyl (766 cm^−1^, 1015 cm^−1^, 1306 cm^−1^ and 1428 cm^−1^) were severely overlapped with the spectral bands of Rhodamine 6G and 4-mercaptopyridine. Meanwhile, traditional RRs were interfered by food samples, especially soybeans and grapes. By contrast, 4-TEAE possessed a highly distinguished and stable peak at 1998 cm^−1^, which gifted 4-TEAE superior anti-interference ability to the traditional RRs.

In summary, anti-interference SERS tags were successfully synthesized. To test the stability of anti-interference SERS tags, three portions of freshly prepared 4-TEAE/Au NPs solution were detected by Raman spectrometer at the same time every day for 10 days. As shown in Figure 2f, the anti-interference SERS tags presented a slight fluctuation in the characteristic peak (1998 cm^−1^), with a Raman intensity deviation (RID) of less than 6%. Thus, the proposed anti-interference SERS tags were stable enough for OTA detection.

### 3.2. Characterization of SERS Probe and Capture Probe

#### 3.2.1. Characterization of SERS Probe

SERS tags link with special single-stranded DNA (OTA-aptamer), which can specifically interact with OTA, gifting SERS probes with specific recognition ability to OTA. To achieve this, 4-TEAE/Au NPs was coated with aptamer via Au-S bond. As shown in Figure 3b, the EDX spectroscopy of 4-TEAE/Au NPs presented multiple peaks, which represented Au, C, N and S elements, respectively. Therefore, the SERS probes with specific recognition of OTA were successfully produced. The result of UV-vis spectra also confirmed the preparation of SERS probes. As shown in Figure 3a, the maximum absorption of Au was red shift from 526 nm to 530 nm, while 4-TEAE/Au NPs were linked with OTA-aptamers.

#### 3.2.2. Characterization of Capture Probes

Fe_3_O_4_ NPs were selected as adsorption substrate, due to the large surface area, and it was easily separated and aggregated by external magnet. Furthermore, highly stable and biocompatible Fe_3_O_4_ NPs combined with cApt to form efficient and sensitive capture probes. TEM was applied to describe the form and size of Fe_3_O_4_ NPs. As shown in Appendix A, the Fe_3_O_4_ NPs are spherical, with an average size of 20 nm. Meanwhile, EDX spectroscopy provided sufficient proof for cApt binding to Fe_3_O_4_ NPs. As shown in Figure 3c,d, the characteristic peaks of Fe and O appeared in the spectrum of Fe_3_O_4_ NPs sample, while the characteristic peaks of Fe, O, C and P were observed in the spectra of Fe_3_O_4_ NPs/cApt. Thus, the capture probe with strong separation ability and high specificity was successfully fabricated.

### 3.3. Optimization of Experiment Conditions

The conditions of probes synthesis and competitive reaction conditions are vital to the sensitivity and specificity of aptasensor. Five key factors of aptasensor (concentration of 4-TEAE, concentration of aptamer, concentration of cApt, pH, and time of competitive reaction) were optimized to the best performance of OTA detection.

#### 3.3.1. Optimization of Probe Synthesis

Firstly, the concentrations of 4-TEAE were altered from 10 to 80 μM, and the Raman intensities of different 4-TEAE concentrations at 1998 cm^−1^ were obtained by counting the results of three experiments. As shown in Figure 4a, the Raman intensity of 4-TEAE at 1998 cm^−1^ increased to the maximum when the concentration of 4-TEAE increased from 10 μM to 50 μM, while it reduced as the concentration of 4-TEAE increased to 80 μM. It could be clearly distinguished in Figure 4a, when the concentration of 4-TEAE reached 50 nM, the Raman intensity at 1998 cm^−1^ was much higher than that of other concentrations. Thus, the optimum concentration of 4-TEAE was selected as 50 nM.

To investigate the effect of aptamer concentration on aptsensor performance, SERS probes with different concentrations (100–500 nM) of aptamers were combined with capture probes and separated by external magnets. Three parallel SERS intensities among different aptamer concentrations (100–500 nM) were counted and compared. In Figure 4b, the Raman intensity at 1998 cm^−1^ increased when the concentration of aptamer increased in the range of 100–300 nM, while it declined as the concentration of aptamer increased from 300 to 500 nM. When the level of aptamer was too little or reached excessive levels, the aptasensor performances were much lower than that at 300 nM. Therefore, the optimum aptamer concentration was 300 nM.

Finally, the optimal load of cApt to capture probes was analyzed by comparing the SERS performances of capture probes with different cApt concentrations. As depicted in Figure 4c, the Raman intensity continuously enhanced until cApt concentration increased to 300 nM, while it decreased as the cApt concentration increased to 500 nM. Hence, the optimal concentration of cApt was set as 300 nM.

#### 3.3.2. Optimization of Competitive Reaction Conditions

The time of competitive reaction has an important effect on aptasensor performance. In order to get the optimal reaction time, a series of reaction times (5, 10, 15, 20, 25, 30 min) were selected to determine the best reaction time. As shown in Figure 4d, with the increase of time, more SERS probes were separated from capture probes due to the presence of OTA. The Raman intensity of SERS probe showed a downward trend, and the Raman intensity was stable at 25 min, which was recorded as the optimal reaction time.

pH and temperature are the other two key factors of the competitive reaction, which, in turn, become the key factors affecting the sensitivity of the aptasensor. As shown in Appendix A, with the increase of OH^−^, more bindings of SERS probes and capture probes were destroyed by OTA. Raman intensity declined, and reached the minimum at the concentration of H^+^ was 10^−7^ M. In addition, owing to the high temperature inflicting damage to probes, referring to the relevant experiments [27,28], we determined that the competitive reaction should be carried out at 37 °C. Hence, a pH value of 7 and a temperature of 37 °C were the optimal conditions for the competitive reaction.

### 3.4. Analytical Performance of the Aptasensor

The aptasensor was performed for OTA detection under the optimal conditions, and the Raman responses to different concentrations of OTA were measured three times. Figure 5a depicted the negative correlation between Raman intensity and OTA concentration. In Figure 5c, the Raman response of 4-TEAE decreased with the concentration of OTA increasing from 0 to 40 nM. Figure 5b showed that the Raman intensity at 1998 cm^−1^ possessed a good linear relationship with the OTA concentration in the range of 0.1–40 nM. The correction curve is y = 2152.14 − 46.73x, R^2^ = 0.991, with the LOD of 30 pM.

### 3.5. Quantification of OTA in Real Samples

To validate the practicability of the aptasensor, OTA detection was performed in real samples (soybean, grape and milk) for the recovery rate. In Table 1, different concentrations of OTA were added to those samples; the samples were pre-treated according to Section 2.6 and a recovery assay was carried out. The recovery rates of the OTA in these samples were found to be 83.3–100.8%, with relative standard deviations (RSDs) of 1.5–8.3%. The obtained results were in good agreement with those determined by the HPLC-MS/MS method, indicating that the aptasensor presents great practicability and reliability for OTA detection in real samples.

### 3.6. Selectivity

In this work, the specificity of the aptasensor was verified by comparing the Raman responds of 300 nM of mycotoxins (AFB1, AFM1, ZEN, DON and TMAE2) and OTA (30 nM) under the same conditions. As shown in Figure 6, the Raman responds of the mycotoxins at 1998 cm^−1^ were significantly higher than that of OTA, indicating the aptasensor possessed a marked affinity to OTA.

The performance of our aptasensor were also compared to those of previously proposed analysis methods [29,30,31,32,33,34]. Appendix A revealed that the proposed aptasensor for OTA detection was a novel, practical, and accurate method.

## 4. Conclusions

In this study, a novel anti-interference aptasensor was developed for OTA detection using 4-TEAE/Au NPs/Apt as SERS probes, and Fe_3_O_4_ NPs/cApt as capture probes. The RRs (4-TEAE) possessed a strong and stable Raman scattering at 1998 cm^−1^, effectively avoiding the interference of the biomolecules in food. The specific interaction of OTA and aptamer provided high specificity and sensitivity for aptamer biosensor. The response of the sensor to OTA can be obtained through a miniature handheld Raman spectrometer, which has flexible application and avoids the limitations of complex operating conditions. Under the optimal conditions, the developed aptasensor presented a linear detection range of 0.1–40 nM, with a LOD of 30 pM to OTA. Aside from this, the results of aptasensor for OTA detection in soybean, grape and milk was also tested, and satisfactory recoveries were obtained. In summary, the anti-interference aptasensor showed great performance in OTA detection and could be applied to enact precise detection of OTA residues in food samples.

## Figures and Tables

**Figure 1 foods-11-03407-f001:**
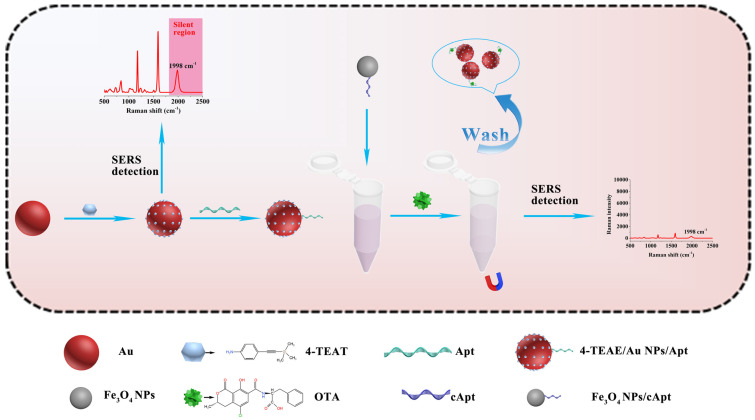
Schematic representation of the principle for the alkyne-mediated SERS aptasensor for anti-interference Ochratoxin A detection.

**Figure 2 foods-11-03407-f002:**
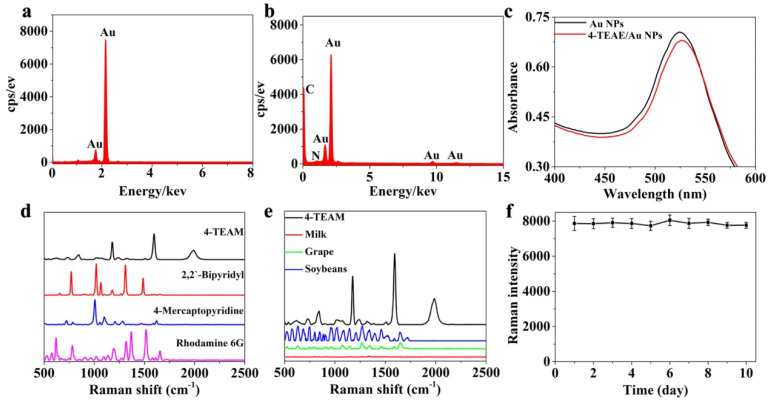
Characterization of the anti-interference SERS tag: (**a**) EDS spectroscopy of Au NPs. (**b**) EDS spectroscopy of 4-TAAE/Au NPs. (**c**) UV−vis spectrum of 4-TEAE/Au NPs (red), Au NPs (black). (**d**) Raman scatterings of conventional RRs and anti-interference SERS probe (black). (**e**) Raman scatterings of food samples and anti-interference SERS tag (black). (**f**) Stability of SERS tag for ten days.

**Figure 3 foods-11-03407-f003:**
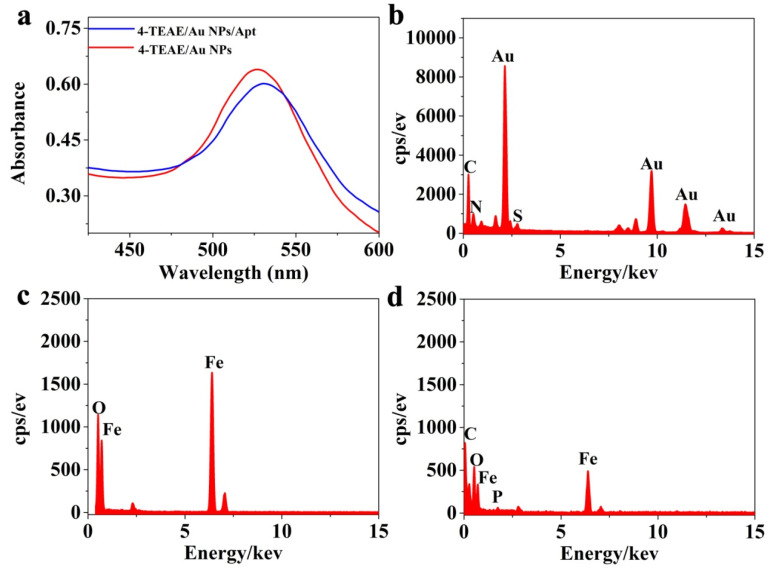
(**a**) UV−vis spectrum of 4-TEAE/Au NPs (red) and 4-TEAE/Au NPs/Apt (blue). EDS spectroscopy of (**b**) 4-TEAE/Au NPs/Apt, (**c**) Fe_3_O_4_ NPs and (**d**) and Fe_3_O_4_ NPs/cApt.

**Figure 4 foods-11-03407-f004:**
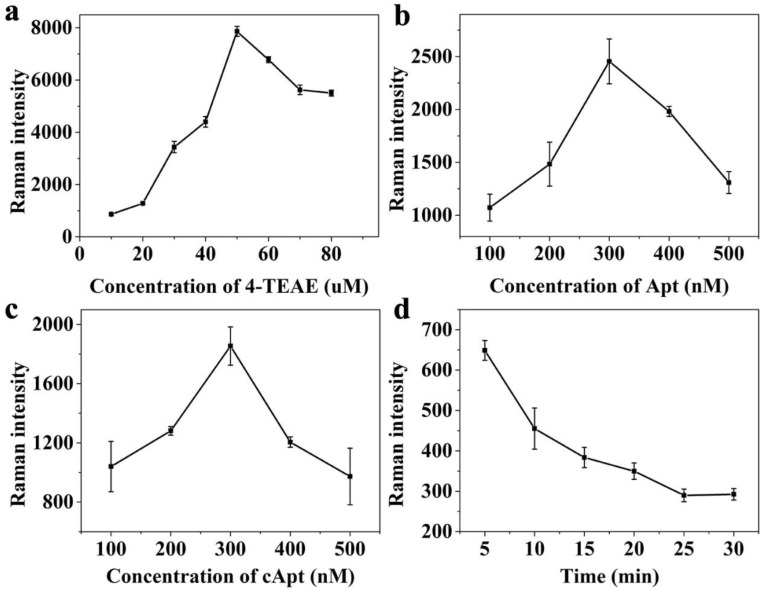
Raman intensity detected at 1998 cm^−1^ for the optimization of (**a**) 4-TEAE concentration, (**b**) Apt concentration, (**c**) cApt concentration and (**d**) competitive reaction time.

**Figure 5 foods-11-03407-f005:**
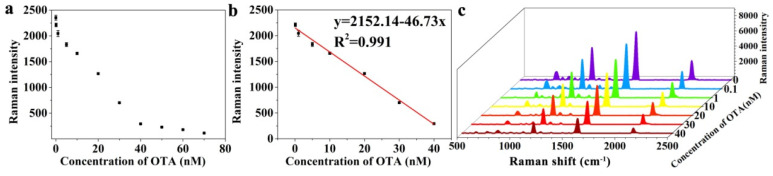
Analytical performance of the aptasensor: (**a**) SERS intensity difference at the peak of 1998 cm^−1^ with the different OTA concentrations, (**b**) linear relationship between SERS intensity and OTA concentration, (**c**) SERS spectra for OTA detection in the concentration range of 0–40 nM.

**Figure 6 foods-11-03407-f006:**
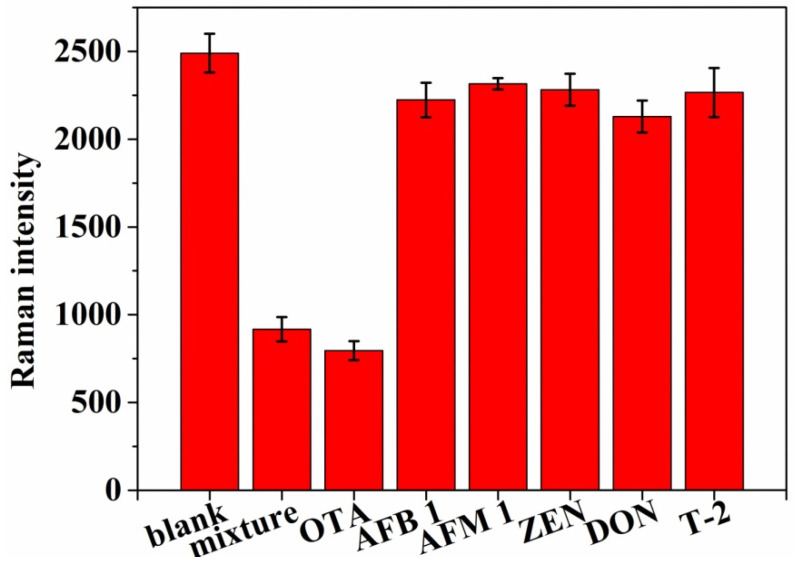
Selectivity evaluation of the ant-interference aptasensor for OTA detection (30 nM) against other mycotoxins (300 nM).

**Table 1 foods-11-03407-t001:** Results of OTA Residues Detection in Real Samples by Aptasensor.

		SERS		HPLC−MS/MS	
Sample	Added (nM)	Found (nM)	Recoveries/RSD (%)	Found (nM)	Recoveries/RSD ^1^ (%)
soybean	1	0.951	95.1/2.1	0.991	99.1/8.1
	5	4.68	93.6/3.5	4.97	99.4/5.2
	10	10.08	100.8/5.3	9.87	98.7/4.1
grape	1	0.997	99.7/2.8	0.981	98.1/5.2
	5	4.56	91.2/1.5	4.86	97.2/5.4
	10	0.954	95.4/4.1	10.2	100.2/4.3
milk	1	0.938	93.8/2.7	1.03	103.0/4.5
	5	4.165	83.3/8.3	5.25	105.4/7.3
	10	9.42	94.2/3.1	10.1	100.1/3.9

^1^ RSDs: relative standard deviation.

## Data Availability

Data are contained within the article.

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
