# Peer review of "An Alkyne-Mediated SERS Aptasensor for Anti-Interference Ochratoxin A Detection in Real Samples"

_foods, 2022, doi:10.3390/foods11213407_

Round 1
Reviewer 1 Report
I have reviewed the manuscript entitled "An Alkyne-mediated SERS Aptasensor for Anti-interference OTA Detection in Real Samples ". This work holds practical application. The authors have efficiently mentioned the background studies in the field, found research gaps and formulated well-defined objectives based on these gaps. Material and methods are well explained. Results are mentioned clearly supported by data in form of the required number of figures and tables. Thus, I recommend minor revision as suggested below.
Line 18, replace “Fe3O4” with “Fe3O4”.
Line 26, mention food samples instead “food”.
Line 30, author may wish to write “naturally occurring mycotoxin”.
Line 100, 105, 131, is it Ultra-pur or Ultra-pure?
Line 133, replace “cm-1” with “cm-1”.
Line 291, mention full form of abbreviations used in Table as footnotes.
Author Response
Dear Reviewer:
Thank you for your letter and for the reviewer's comments concerning our manuscript “An Alkyne-mediated SERS Aptasensor for Anti-interference Ochratoxin A Detection in Real Samples” Those comments are very helpful for us to revise and improve our paper. The main corrections in the paper and the responses to the reviewer's comments are as follows:
Line 18, replace “Fe3O4” with “Fe3O4”.
Response: We appreciate the reviewer’s suggestion and replace “Fe3O4” with “Fe3O4” in line 17.
Line 26, mention food samples instead “food”.
Response: Thank you for your advice. We have substituted “food samples” for “food” in line 26.
Line 30, author may wish to write “naturally occurring mycotoxin”.
Response: Thank you for careful reading. We have substituted “naturally occurring mycotoxin” for “occurring mycotoxin” in line 30.
Line 100, 105, 131, is it Ultra-pur or Ultra-pure?
Response: Thanks for your questions. We have replaced “Ultra-pur” with “Ultra-pure” in lines 107, 112, 119, 124. 129, 132, 159, 165 and 166.
Line 133, replace “cm-1” with “cm-1”.
Response: Thank you for careful reading. We have replaced “cm-1” with “cm-1” in lines 15, 20 and 161.
Line 291, mention full form of abbreviations used in Table as footnotes.
Response: Thank you for careful reading. We have mentioned the full form of “RSDs” in footnotes.
Special thanks to you for your good comments. We tried our best to improve the manuscript and made some changes in the manuscript. These changes will not influence the content and framework of the paper. We appreciate Editors/Reviewers’ warm work earnestly, and hope the correction will meet with approval. Once again, thank you very much for your comments and suggestions.
Sincerely yours,
Ph.D., Mengmeng Yan

Reviewer 2 Report
In this experiment, a novel anti-interference aptasensor was developed for the anti-interference detection of ochratoxin A (OTA), an abundant food-contaminating mycotoxins. The aptasensor could avoid the interference of food biomolecules in surface-enhanced Raman spectroscopy analysis, proved superior anti-interference ability to the traditional Raman reporters, and showed great performance of OTA residue detection in soybean, grape, and milk samples.
The acquired results corresponded with the results determined by HPLC-MS/MS method. The aptasensor revealed pronounced affinity to OTA, thus, it can be useful and reliable for OTA detection in real samples.
The topic is important and the manuscript provides interesting results. I would recommend this manuscript for publication after the following suggestions have been attended to:
Is there any statistical analysis? If yes, please add in Methods and text including figures and tables.
Line 77: sentence should be revised
Lines 100, 105, 114. 117, 122, 125, 131, 137, 138: “Ultra-pure water”
Lines 114, 118, 125, 137: delete “for”
Line 182: “Fig. 2d-e”
Line 191: “Fig. 2f”
Line 303: “was...”
Author Response
Dear Reviewer:
Thank you for your letter and for the reviewer’s comments concerning our manuscript “An Alkyne-mediated SERS Aptasensor for Anti-interference Ochratoxin A Detection in Real Samples” Those comments are very helpful for us to revise and improve our paper. The main corrections in the paper and the responses to the reviewer's comments are as follows:
Is there any statistical analysis? If yes, please add in Methods and text including figures and tables.
Response: Thank you for your advice. We have added 2.8 section as Statistical analysis with the specific content of “All results were obtained from cubic parallel experiments, and standard deviation (SD) were represented with error bars one. Statistical analysis was performed with Origin 9.5. Ultimately, the limit of detection (LOD) was computed as LOD = 3 × SD/blank.” in line 184-187.
Line 77: sentence should be revised
Response: Thank you for your advice. The title of Figure. 1 has been changed to “Schematic representation of the principle for the alkyne-mediated SERS aptasensor for anti-interference Ochratoxin A detection in real samples.” In line 83-84.
Lines 100, 105, 114. 117, 122, 125, 131, 137, 138: “Ultra-pure water”
Response: Thank you for your suggestions. We have replaced “Ultra-pur” with “Ultra-pure” in lines 107, 112, 119, 124. 129, 132, 159, 165 and 166.
Lines 114, 118, 125, 137: delete “for”
Response: Thank you for your suggestions. We have substituted “three times” for “for three times” in lines 119, 124, 132 and 132.
Line 182: “Fig. 2d-e”
Response: Thank you for your amendment to our work. We have replaced “Fig. d-e” with “Fig. 2d-e” in line 215.
Line 191: “Fig. 2f”
Response: Thank you for your amendment to our work. We have replaced “Fig. 1f” with “Fig. 2f” in line 224.
Line 303: “was...”
Response: Thank you for your suggestion. We have replaced “were novel, practical, and accurate method” with “was novel, practical, and accurate method” in line 339.
Special thanks to you for your good comments. We tried our best to improve the manuscript and made some changes in the manuscript. These changes will not influence the content and framework of the paper. We appreciate Editors/Reviewers’ warm work earnestly, and hope the correction will meet with approval. Once again, thank you very much for your comments and suggestions.
Sincerely yours,
Ph.D., Mengmeng Yan
Reviewer 3 Report
The manuscript is well written and concise, but it is necessery to add significancy of investigation.
Author Response
Dear Reviewer:
Thank you for your letter and for the reviewer’s comments concerning our manuscript “An Alkyne-mediated SERS Aptasensor for Anti-interference Ochratoxin A Detection in Real Samples” Those comments are very helpful for us to revise and improve our paper. The main corrections in the paper and the responses to the reviewer's comments are as follows:
The manuscript is well written and concise, but it is necessery to add significancy of investigation.
Response: Thank you for your advice. We have added the significancy of our investigation with the specific content of “As a major classification of anti-interference RRs, alkynyl-containing RRs have unique Raman shift in the silent region. The exploration of alkynyl-containing RRs is helpful to the development of Raman silent region tags and expand the application potential of anti-interference RRs. But so far, there are no reports about applying alkynyl compound as anti-interference RRs in the food safety field.” in line 67-70.
Special thanks to you for your good comments. We tried our best to improve the manuscript and made some changes in the manuscript. These changes will not influence the content and framework of the paper. We appreciate Editors/Reviewers' warm work earnestly, and hope the correction will meet with approval. Once again, thank you very much for your comments and suggestions.
Sincerely yours,
Ph.D., Mengmeng Yan

Reviewer 4 Report
In this manuscript, authors developed SERS aptasensor to detect OTA. They fabricated SERS probes and capture probes conjugated with aptamers and cApt and analyzed them. Then, the condition for the measurements were optimized and OTA was detected using the developed biosensor. The topic of this manuscript is well fit to the Foods and the logic of manuscript is scientifically sound. There are comments for the modification as below:
1. In title, authors used abbreviations. Especially, OTA is not familiar with readers.
2. In section 2.1, line 81, HAuCl4 is not gold chloride but chloroauric acid.
3. In figure1, 4-TEAT looks like a typo. In addition, 4-TEAE is not described in section 2.1.
4. In section 2.3, authors treated Tris (2-carboxyethy1) phosphine (TCEP) to obtain 4-TEAE/Au NPs. Furthermore, OTA aptamer was mixed with TCEP solution before addition to Au NPs and OTA aptamer was additionally added after aptamer/TCEP solution treatment (line 111 – 116). It is confusing and not clear. Check the protocol carefully and describe clearly.
5. Authors compared the results from fabricated sensor with existing HPLC-MS/MS method. The values of recoveries from HPLC-MS/MS seems better than the sensor results. What is the novelty of developed method?
Author Response
Dear Reviewer:
Thank you for your letter and for the reviewer’s comments concerning our manuscript “An Alkyne-mediated SERS Aptasensor for Anti-interference Ochratoxin A Detection in Real Samples” Those comments are very helpful for us to revise and improve our paper. The main corrections in the paper and the responses to the reviewer's comments are as follows:
In title, authors used abbreviations. Especially, OTA is not familia with readers.
Response: Thank you for your advice. The title has been changed to “An Alkyne-mediated SERS Aptasensor for Anti-interference Ochratoxin A Detection in Real Samples”.
In section 2.1, line 81, HAuCl4 is not gold chloride but chloroauric acid.
Response: Thank you for careful reading. We have replaced “gold chloride” with “chloroauric acid” in section 2.1, line 88.
In figure. 1, 4-TEAT looks like a typo. In addition, 4-TEAE is not described in section 2.1.
Response: Thank you for careful reading. We have added 4-[(Trimethylsilyl) ethynyl] aniline as the full form for 4-TEAE in line 74, and replaced “4-[(Trimethylsilyl) ethynyl] aniline” with “4-TEAE” in part 2.1.
In section 2.3, authors treated Tris (2-carboxyethy1) phosphine (TCEP) to obtain 4-TEAE/Au NPs. Furthermore, OTA aptamer was mixed with TCEP solution before addition to Au NPs and OTA aptamer was additionally added after aptamer/TCEP solution treatment (line 111 – 116). It is confusing and not clear. Check the protocol carefully and describe clearly.
Response: Thank you for your comment. In this work, Tris (2-carboxyethy1) phosphine (TCEP) served as the solvent of aptamer to protect the 5’-HS of aptamer. In section 2.3, line117-123, “and the Au NPs solution was stored at 4 ℃ for further use.
OTA aptamer was dissolved to 100 μM with TCEP solution and then diluted to 10 μM with PBS solution for further use. The 0.87 mL Au NPs solution was mixed with 100 μL 4-TEAE in a scroll oscillator for 1 hour. The resulted solution was centrifuged under 6000 rpm for 15 minutes and washed with ultra-pure water three times to obtain 4-TEAE fixed Au NPs (Au NPs-4-TEAE). Then, 30 μL (10 μM) OTA aptamer was co-incubated with Au NPs-4-TEAE solution at room temperature for 4 h.” has been changed to “The 1 mL of cooled Au NPs solution was mixed with 100 μL 4-TEAE in a scroll oscillator for 1 hour. The resulted solution was centrifuged under 6000 rpm for 15 minutes and washed with ultra-pure water three times to obtain 4-TEAE fixed Au NPs (Au NPs-4-TEAE).
OTA aptamer was dissolved to 100 μM with TCEP solution and then diluted to 10 μM with PBS solution. Then, 30 μL (10 μM) OTA aptamer was co-incubated with Au NPs-4-TEAE solution at room temperature for 4 h.”
Authors compared the results from fabricated sensor with existing HPLC-MS/MS method. The values of recoveries from HPLC-MS/MS seems better than the sensor results. What is the novelty of developed method?
Response: Thank you for your comment. Our aptasensor is the first exploration of using alkynyl-containing RRs in food safety field, which is an important supplement to the application of anti-interference RRs (line 67-71). Meanwhile, the response of aptasensor to OTA can be obtained through a miniature handheld Raman spectrometer, which has the flexible application scenario and escapes the limitations of complex operating conditions (line 331-334).
Special thanks to you for your good comments. We tried our best to improve the manuscript and made some changes in the manuscript. These changes will not influence the content and framework of the paper. We appreciate Editors/Reviewers’ warm work earnestly, and hope the correction will meet with approval. Once again, thank you very much for your comments and suggestions.
Sincerely yours,
Ph.D., Mengmeng Yan
